# TESTING CROSS-LINGUAL TEXT COMPREHENSION IN LLMS USING NEXT SENTENCE PREDICTION.

## ABSTRACT

While large language models are trained on massive datasets, this data is heavily skewed towards English. Does their impressive performance reflect genuine ability or just this data advantage? To find out, we tested them in a setting where they couldn't rely on data abundance: low-resource languages. Building on prior work Agarwal et al. (2025) that used Next Sentence Prediction (NSP) as a test, we created a large-scale benchmark with 10,000 questions each for English (a high-resource language), Swahili (medium-resource), and Hausa (low-resource). We then tested several top models, including GPT-4 Turbo, Gemini 1.5 Flash, and LLaMA 3 70B, to see how their performance holds up. The results painted a clear picture of how levels of language resources impact outcomes. While all models excelled in English, their accuracy dropped in Swahili and fell sharply in Hausa, with LLaMA 3 struggling the most. The story became even more interesting when we introduced Chain-of-Thought (CoT) prompting. For the struggling LLaMA 3, CoT acted as a helpful guide, significantly boosting its accuracy. However, for the more capable GPT-4 and Gemini, the same technique often backfired, leading to a kind of "overthinking" that hurt their results in the cross-lingual context. This reveals that Chain-of-Thought is not a universal solution; its effectiveness depends heavily on the model's baseline capability and the specific context of the task. Our framework pinpoints LLM weaknesses, highlights when CoT helps or hinders cross-lingual NSP performance, and factors influencing their decisions.

## 1 INTRODUCTION

A key challenge in education, especially in developing countries, is creating effective tools to teach reading comprehension. Platforms like the RoboTutor tablet Carnegie Mellon University (2023) need scalable ways to automatically check if a child is following a story, a task that is difficult to do across many languages without manual effort. Our solution is to use a simple task called "Next Sentence Prediction" (NSP). After reading a story passage, the student simply picks the sentence that should come next, testing their ability to understand how sentences connect to form a coherent story. For example:

> Given the following story context:
>
> *Chicken and Millipede were friends. They went to play football. Chicken was the goal keeper.*
>
> Which sentence comes next?
>
> **A:** *Millipede scored three goals.* (The correct, logical next sentence)
>
> **B:** *Chicken swallowed Millipede.* (A sentence from later in the story that doesn't fit here)

While others have used this idea, our work goes further by testing if an AI's comprehension is genuine or just a result of seeing massive amounts of English text. To do this, we built a large,

balanced benchmark with 10,000 questions each for English (high-resource), Swahili (medium-resource), and Hausa (low-resource). We tested several popular models, including GPT-4 Turbo, Gemini 1.5 Flash, and LLaMA 3 70B, to see how they compare.

To understand what influences a model's decisions, we analyzed factors like context_length, distractor_distance, semantic_similarity, and perplexity. We used a logistic regression model to find which factors most accurately predicted a correct answer. We also explored Chain-of-Thought (CoT) prompting, where the model explains its reasoning, to see if it improves accuracy and if its explanations could be used as a teaching tool for students. In summary, this work makes the following key contributions:

- A large, diverse, and equal dataset for testing text comprehension across three languages with varying resource levels: English, Swahili, and Hausa.
- A benchmark that evaluates and compares how well top-tier language models, including GPT-4 turbo, Gemini 1.5 Flash, and LLaMA 3 70B perform on this cross-lingual comprehension task with and without COT prompting.
- A novel proposal for using Chain-of-Thought (CoT) explanations as a real-time teaching tool in educational platforms like RoboTutor to help children understand their reading mistakes.

## 2 RELATED WORK

### 2.1 EVALUATING TEXT COMPREHENSION IN LLMS

The evaluation of Large Language Model (LLM) comprehension often relies on broad benchmarks like GLUE Wang et al. (2018) and SuperGLUE Wang et al. (2019), or on tests for specific skills like question-answering with SQuAD Rajpurkar et al. (2016) and commonsense inference with HellaSwag Zellers et al. (2019). While vital for measuring AI's progress, these benchmarks are often too complex for simple and scalable use in practical applications like educational apps.

### 2.2 NEXT SENTENCE PREDICTION FOR COMPREHENSION EVALUATION

Our work instead uses the Next Sentence Prediction (NSP) task, first introduced as a pre-training objective for BERT Devlin et al. (2019). Although it was later deprecated in models like RoBERTa Liu et al. (2019) because models learned to use superficial cues, Agarwal et al. (2025) recently revisited NSP as a direct benchmark for evaluating ChatGPT's grasp of narrative flow. However, this foundational approach had significant limitations, including its focus on a single model, a narrow linguistic scope with a severe data imbalance between English and Swahili, and an analysis that relied on surface-level features. It also did not explore advanced prompting techniques like Chain-of-Thought Wei et al. (2022). Our research conducts a more ambitious and rigorous investigation to address these gaps. We introduce a large, balanced dataset with 10,000 questions each for English, Swahili, and Hausa. We expand the evaluation to a suite of models including GPT-4 Turbo, Gemini 1.5 Flash, and Llama 3 70B, and we incorporate deeper analytical features like perplexity and semantic similarity. Finally, we are the first to systematically apply and analyze CoT prompting for this task, proposing its use as a pedagogical tool in RoboTutor.

## 3 DATASET

To evaluate cross-lingual comprehension, we constructed a new, large-scale Next Sentence Prediction (NSP) dataset. The process involved four stages: scraping stories, generating questions, enriching the data with deep-learning features, and validating the final dataset.

### 3.1 SCRAPING

Our data source was the African Storybooks website Saide (2025), an open-access digital library of children's stories. This source was ideal due to its strong narrative structure, essential for meaningful NSP questions, and its rich repository of texts in our target languages. We developed a script

to download approximately 400 storybooks each for English, Swahili, and Hausa. We extracted text from the EPUB format, which proved cleaner than PDF, converted the files to plain text, and concatenated them into a single master file for each language.

## 3.2 Generating NSP Questions

We used a custom Python script to generate 10,000 NSP questions for each language. The script segments each story into sentences and uses a sliding window to define a context (3-10 sentences) and a true answer (the next sentence). A distractor is then randomly selected from a later part of the story, with its distractor_distance varied from 2 to 10 sentences away. The script records these features, randomizes the option order, and saves the final question to a CSV file.

## 3.3 New Features

To enable deeper analysis, we enriched the dataset with two features:

- **Semantic Similarity:** To measure how close in meaning an option is to the context, we generated text embeddings using the sentence-transformers/paraphrase-multilingual-MiniLM-L12-v2 model Reimers & Gurevych (2019) and calculated the cosine similarity between the context and each option.

- **Perplexity (PPL):** This measures how "surprising" a sentence is to a language model given the context. After experimenting to find models that produced reliable scores, we chose language-specific models: mistralai/Mistral-7B-Instruct-v0.3 Jiang et al. (2023) for English, Jacaranda/UlizaLlama3 for Swahili, and Jacaranda/HausaLlama for Hausa. To isolate the surprisal of the option sentence alone, we calculated the perplexity on the combined context + option sequence but masked the context tokens from the loss calculation.

## 3.4 Validating Dataset

A critical final step was to validate that our features behaved as hypothesized. As shown in Figure 7 (Appendix), we confirmed that as the distractor distance increases, its semantic similarity to the context decreases while its perplexity generally increases. This validation gives us confidence that our dataset is a reliable tool for analyzing model comprehension.

## 4 Evaluation

This section details our experimental setup and presents the core results from our cross-lingual benchmark. The findings reveal a fascinating picture of how today's top AI models comprehend stories in languages with varying levels of data resources.

## 4.1 Methodology

To ensure a fair and clear assessment, we established a consistent methodology. We selected three leading LLMs to get a wide view of the field:

**GPT-4 Turbo** Achiam et al. (2023), **Gemini 1.5 Flash** Team et al. (2024), and **Llama 3 70B Instruct** Meta AI (2024). Each model was presented with the NSP questions from our English, Swahili, and Hausa datasets. We used two prompting strategies:

- **Direct Answering (Baseline):** The model was prompted to return only the letter of its choice (A or B). This was tested on the full dataset of 10,000 questions per language.

- **Chain-of-Thought (CoT):** The model was instructed to explain its reasoning before providing an answer. Due to computational costs, this was performed on a randomly selected 1,000-question subsample for each language, with the baseline prompt also run on this same sample for a direct comparison.

  Our primary metric is **accuracy**, the percentage of questions answered correctly.

## 4.2 CROSS-LINGUAL BENCHMARK PERFORMANCE

Running our baseline "Direct Answering" tests on the full 10,000-question datasets, a clear story emerged.

| Models | English Accuracy | Swahili Accuracy | Hausa Accuracy |
|---|---|---|---|
| GPT - 4 Turbo | 81.61% | 78.35% | 76.02% |
| Gemini 1.5 Flash | 80.79% | 77.13% | 75.04% |
| LLaMA 3 70B | 80.71% | 68.71% | 59.43% |

Table 1: Benchmark on 10,000 NSP questions (Comparison of model accuracy on NSP questions)

The baseline results, shown in Table 1, reveal that a model's comprehension is deeply connected to the language's resource level. In the high-resource English setting, all three models performed impressively, with accuracies packed into a tight ∼1% range. The picture changed in Swahili, where GPT-4 and Gemini's performance dipped only slightly, but LLaMA 3's accuracy dropped a full 12 percentage points. This gap became a chasm in the low-resource Hausa setting. While GPT-4 and Gemini remained respectable at over 75% accuracy, LLaMA 3's performance plummeted to 59.43%, barely better than a random guess. This starkly suggests that its ability to model sentence connections is severely weakened in data-scarce environments.

## 4.3 IMPACT OF CHAIN-OF-THOUGHT (COT)

| Language | Model | Baseline Accuracy (1k) | COT Accuracy (1k) | % Improvement |
|---|---|---|---|---|
| English | GPT - 4 | 82.50% | 83.00% | +0.5% |
| | Gemini | 82.70% | 80.20% | -2.5% |
| | LLaMA 3 | 82.60% | 82.30% | -0.3% |
| Swahili | GPT - 4 | 77.60% | 74.50% | -3.1% |
| | Gemini | 77.20% | 71.10% | -6.1% |
| | LLaMA 3 | 68.40% | 73.00% | +4.6% |
| Hausa | GPT - 4 | 76.20% | 75.90% | -0.3% |
| | Gemini | 74.40% | 71.30% | -3.1% |
| | LLaMA 3 | 60.60% | 65.40% | +4.8% |

Table 2: Benchmark on 1,000 NSP questions with COT (Comparison of model accuracy with and without Chain-of-Thought prompting.)

A central question was whether CoT prompting could improve comprehension in these challenging settings. The results, presented in Table 2, show a complex and often contradictory effect.

- **CoT as a Guide:** For LLaMA 3, the model that struggled most, CoT provided a significant benefit. In the low-resource contexts of Swahili and Hausa, forcing it to reason step-by-step improved its accuracy by a substantial 4.6 and 4.8 percentage points, respectively. This suggests CoT acts as a "reasoning guide" for a model operating at the edge of its capability.

- **CoT as a Hindrance:** Counter-intuitively, for the higher-performing GPT-4 and Gemini, CoT was often detrimental in lower-resource languages. In Swahili, Gemini's accuracy dropped by over 6 percentage points. This suggests an "overthinking" problem; for models with a strong intuitive grasp, the task of generating reasoning may introduce cognitive friction or cause them to justify an initially incorrect impulse.

In summary, the effectiveness of CoT is not universal. It is a powerful tool for guiding weaker models but can be a counter-productive constraint for more capable ones.

## 4.4 ANALYSIS OF INFLUENTIAL FACTORS IN DECISION

To understand what influenced model decisions, we used a logistic regression model, which we found to be the most effective predictor. We analyzed which of our features was the single most

| Language | Model | Most Influential Factor | Interpretation of Factor |
|---|---|---|---|
| English | GPT - 4 | distractor_distance | Farther distractors are easier. |
| | Gemini | distractor_distance | Farther distractors are easier. |
| | LLaMA 3 | distractor_distance | Farther distractors are easier. |
| Swahili | GPT - 4 | distractor_distance | Farther distractors are easier. |
| | Gemini | distractor_distance | Farther distractors are easier. |
| | LLaMA 3 | distractor_length | Longer distractors are harder. |
| Hausa | GPT - 4 | distractor_distance | Farther distractors are easier. |
| | Gemini | distractor_length | Longer distractors are harder. |
| | LLama 3 | distractor_length | Longer distractors are harder. |

Table 3: Decision Factors
(The most predictive factor for a correct answer)

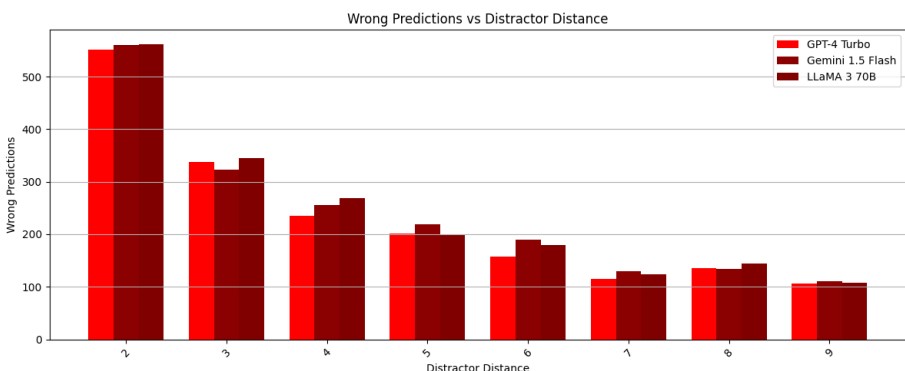

Figure 1: Factor 1 influencing error (Wrong Predictions vs Distractor Distance)

powerful predictor of a correct answer for each LLM. The results, summarized in Table 3, reveal a fascinating shift in strategy as models move from high- to low-resource languages.

In the high-resource English setting, distractor_distance was uniformly the most important factor. This indicates all models used a sophisticated strategy of evaluating logical coherence, which is easier when the distractor is from a distant, unrelated part of the story. This pattern changes in lower-resource languages. In Swahili, LLaMA 3's strategy shifts to rely on distractor_length, suggesting it is confused by longer sentences. This shift is more pronounced in Hausa, where both Gemini and LLaMA 3 show distractor_length as the most influential factor. Only GPT-4 consistently relies on the more sophisticated distractor_distance signal across all languages. This suggests that as the linguistic challenge increases, models may abandon deeper coherence analysis in favor of simpler, surface-level heuristics.

## 5 ERROR ANALYSIS

### 5.1 ANALYSIS OF FACTORS INFLUENCING ERRORS

To understand why models made mistakes, we analyzed the specific factors that made the Next Sentence Prediction task harder. By examining variables like distractor placement and our engineered metrics, we identified the exact situations where the models' ability to understand stories breaks down.

### 5.1.1 ANALYSIS OF FACTORS INFLUENCING ERRORS

Our analysis (Figure 1) shows that models struggled most with "near-miss" distractors; wrong answers drawn from a nearby part of the story (distractor_distance). These are difficult because they are often on the same topic and use similar words, making them hard to distinguish from the truly logical next sentence.

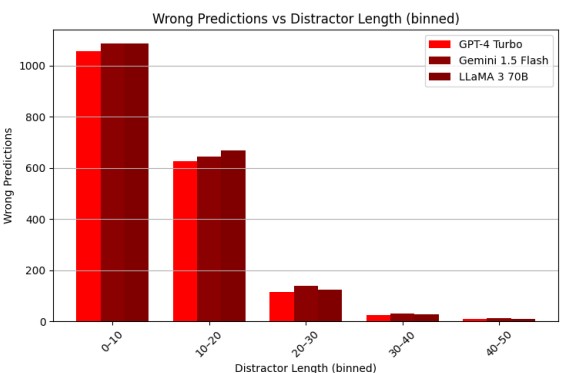

Figure 2: Factor 2 influencing error (Wrong Predictions vs Distractor Length)

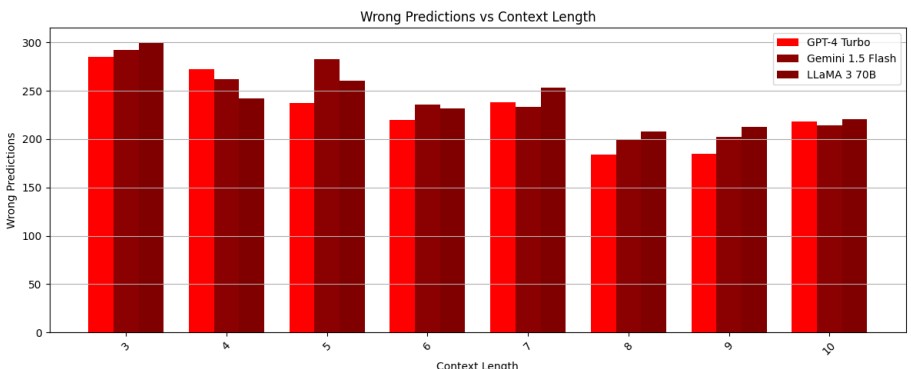

Figure 3: Factor 3 influencing error (Wrong Predictions vs Context Length)

### 5.1.2 THE IMPACT OF DISTRACTORS LENGTH

In a surprising twist, we found (Figure 2) that shorter wrong answers (distractor_length) were better at fooling the models. This is likely because their simple structure and fewer words offer the model fewer textual cues that might give away their incorrectness

### 5.1.3 THE IMPACT OF CONTEXT LENGTH

As expected (Figure 3), providing more story context (context_length) helped reduce errors. However, this effect was less pronounced than other factors, suggesting that simply adding more context is not a magic fix for the most difficult questions .

### 5.2 THE "ZONE OF AMBIGUITY": DELTA PERPLEXITY AND SEMANTIC SIMILARITY

Perhaps our most revealing insight is the "Zone of Ambiguity," a state where error rates for all models peaked when the difference (delta) between the two options' perplexity and semantic similarity scores was close to zero.

- **Delta Perplexity:** (Figure 4) Errors were most frequent when both options were similarly plausible from a probabilistic standpoint. When models could not rely on a clear statistical signal to identify the less "surprising" sentence, they lacked the deeper, causal reasoning needed to make the correct choice.

- **Delta Semantic Similarity:** (Figure 5) We saw the exact same pattern for semantic similarity. When both options were an equally good topical match for the context, the models got confused because they could not use a simple relevance hint to find the right answer.

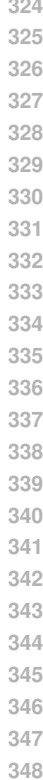
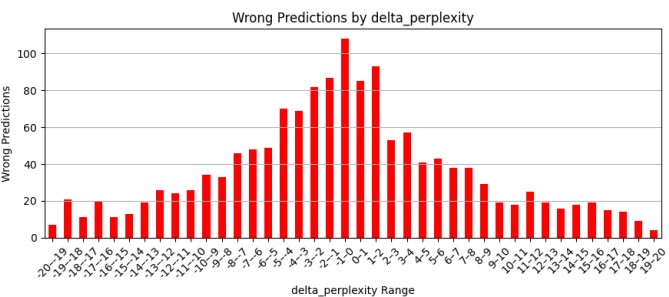

Figure 4: Factor 4 influencing error (Wrong Predictions vs Delta Perplexity)

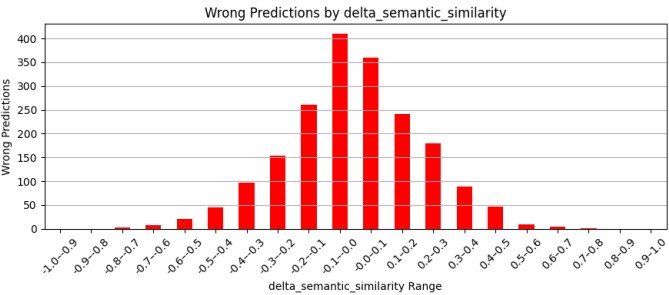

Figure 5: Factor 5 influencing error (Wrong Predictions vs Delta Semantic Similarity)

Together, these findings show that the models are at their weakest when they cannot rely on a clear statistical signal. Their failure in these ambiguous situations demonstrates that they still lean on surface-level heuristics instead of a robust, human-like understanding of story flow.

## 5.3 COHERENCE ERRORS

The most fascinating errors were those that defied our quantitative analysis, where all models failed despite strong statistical signals pointing to the correct answer. These mistakes reveal deep-seated biases in how models process narratives.

**Case Study: When a Story Turns into a Conversation**

In this English example, all three models failed, even though our feature analysis showed Option B was a much better fit (higher semantic similarity, lower perplexity) .

> **Context:** "That causes an upward pressure or force. The upward force holds an airplane in the sky. The closest thing I can do to flying, is in an airplane."
> - **A (Incorrectly Chosen):** "We are speeding along the runway."
> - **B (Correct):** "Have you ever been in an airplane?"

**Analysis of Failure:** The context subtly shifts from a technical explanation to a personal reflection. The logical next step is to continue this new, conversational tone, which Option B does perfectly by asking the reader a direct question . Option A, in contrast, jarringly starts a new, unrelated action scene . The models failed because their fundamental design as autoregressive pattern-matchers defaulted to continuing a descriptive story—the most common pattern in their training data. They were so conditioned to be storytellers that they failed to notice the narrative had cleverly changed into a conversation, ignoring the strong perplexity and semantic cues .

## 5.4 LOW-RESOURCE LANGUAGE FAILURES

- **Mistranslation:** Reasoning based on a completely incorrect translation of an option.

- **Ignoring Narrative Patterns:** Failing to recognize and continue a clear, repetitive pattern established in the story.
- **Logical Leaps & Flawed Temporal Reasoning:** Making assumptions not supported by the context or failing to process the correct sequence of events.

Below case studies are translated for readers understanding:

**Case Study 1: Failure due to Mistranslation**

---

**Context:** A fed-up rabbit (Zomo) decides to leave his lazy hyena friend (Kura).

- **A (LLaMA's Choice):** "He climbed and gathered a lot for them."

- **B (Correct):** "One day, the hyena woke up early in the morning."

---

**Analysis:** LLaMA 3 chose A because it completely mistranslated the sentence as "He got up and started praying a lot." It then invented a plausible but fictional reason about the rabbit seeking "divine intervention." This shows that without basic comprehension of the language, its reasoning is built on a fantasy, leading to a confidently wrong answer.

**Case Study 2: Failure to Follow a Narrative Pattern**

---

**Context:** Fati sneakily takes two pieces of meat from a pot while her mother is distracted.

- **A (Correct):** "Quickly, Fati took a third piece of meat from the pot and ate it."

- **B (LLaMA's Choice):** "She took the spoon from Fati's hand and began to stir the soup."

---

**Analysis:** LLaMA's own reasoning confesses its error: it dismissed the correct answer as "repetitive," choosing the wrong one because it introduced a "new development." It failed because it prioritized novelty over logical coherence, ignoring clear data and narrative signals.

**Case Study 3: Failure in Temporal Logic**

---

**Context:** Fati's mother leaves, promising to sing a special song on return.

- **A (Correct):** "Come out quickly, here is your food!" (This continues the mother's call)

- **B (LLaMA's Choice):** "Fati opened the door and ate her delicious food."

---

**Analysis:** This case reveals a cascade of failures. The model makes a temporal logic error because of a deeper failure: a complete mistranslation of the correct answer. This fundamental comprehension error meant it was easily fooled by misleading statistics, showing how one error can trigger a chain reaction in reasoning.

## 5.5 CHAIN OF THOUGHT FAILURES

Our most surprising result was that for our best models, GPT-4 and Gemini, using Chain-of-Thought (CoT) often degraded their performance in low-resource languages. We found a clear pattern of "overthinking," where a model that answered correctly at baseline would, when asked to reason, ignore a simple, logical answer in favor of a more complex but incorrect one.

**Case Study: Overthinking a Simple Pattern**

A clear example of this occurred in our Hausa dataset, where the baseline versions of both GPT-4 and Gemini answered correctly, but both failed when prompted with CoT.

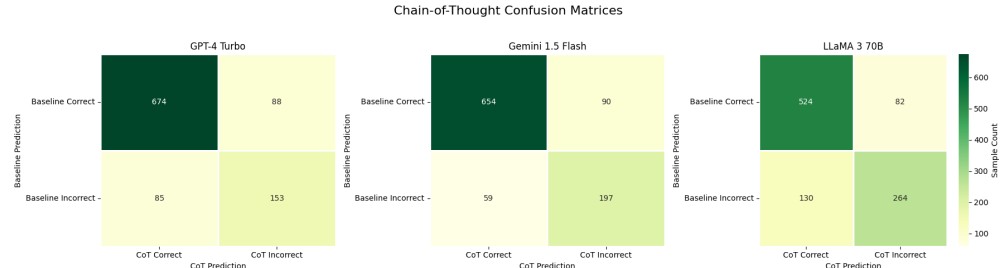

Figure 6: COT confusion (2x2 matrix for correct and incorrect with and without COT predictions)

> **Context (Translated):** "I like milk." "My father likes apples." "I like sweet lemons/oranges."
>
> - **A (Incorrectly Chosen by CoT models):** "All of us like Chin chin (a fried snack)."
>
> - **B (Correct):** "My father likes bread."

**Analysis of the Failure:** The context establishes a simple, alternating pattern of listing individual food preferences. The most logical continuation is Option B, which perfectly maintains this pattern and was correctly identified by the baseline models. However, when asked to reason, the more capable models invented a more complex goal for the text:

- **Gemini's Flawed Reasoning:** It dismissed the correct Option B as "repetitive" and praised the incorrect Option A for "expand[ing] on the established theme in a meaningful way" by introducing a "broader group."
- **GPT-4's Flawed Reasoning:** It similarly argued that Option A was a more "interesting and logical continuation" because it "ties the individual preferences together," thereby "enriching the narrative."

  In both cases, the models failed a simple pattern-matching task because they treated it as a creative writing task. They penalized the correct, logical continuation for being too simple and rewarded the pattern-breaking option for being more "complex" or "meaningful".

# 6 CONCLUSION

In this work, we presented a large-scale, multi-lingual benchmark to test the text comprehension of modern LLMs. Our findings reveal a clear performance gap between high and low-resource languages and uncover the complex, often counter-productive effects of Chain-of-Thought prompting. This final section discusses the limitations of our study and offers concluding remarks on the implications of our work.

## 6.1 LIMITATIONS AND FINAL REMARKS

While our study has limitations regarding its focus on narrative coherence in children's storybooks, a specific set of models, and a subsampled CoT analysis due to computational costs, it provides a critical perspective on cross-lingual AI. Our investigation demonstrates that modern LLM comprehension is fragile and highly dependent on a language's resource level, showing that performance degradation involves not just a drop in accuracy but a strategic shift towards simpler heuristics. Furthermore, we reveal Chain-of-Thought as a double-edged sword that guides weaker models but causes more capable ones to "overthink" and fail. By providing a robust benchmark and a deep analysis of model failures, this work shows that building truly multilingual AI requires looking beyond accuracy to engage with the complex ways these models reason, succeed, and make mistakes. Future work should expand the benchmark across more genres and models, compare model performance directly against human baselines, and explore the application of successful CoT reasoning as a real-time pedagogical tool in educational platforms.

## REPRODUCIBILITY STATEMENT

All datasets, code, and analysis scripts used in this paper are included in the supplementary material, along with a README file describing the file structure and usage. Detailed instructions are provided to reproduce dataset construction, model evaluations, and error analyses. Our methodology (Sections 3–5) and Appendix A include pseudocode and implementation details for dataset generation, feature extraction, and evaluation. All experiments were conducted with fixed random seeds where applicable, and the exact prompts and model settings are documented to enable replication.

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

## A  APPENDIX

### A.1  GENERATING NSP QUESTIONS

This algorithm 1 reads through stories, takes a piece of text (the context), and then creates a multiple-choice question. The question presents two options: the real sentence that comes next in the story and a fake sentence taken from a random, incorrect spot. The goal is to generate a large set of these "Next Sentence Prediction" questions automatically.

---

**Algorithm 1** Generate NSP questions

---

**Input:** master txt file
**Output:** output_file in CSV format with generated questions.
 1: stories = LOAD_STORIES(input_file)
 2: all_questions = []
 3: **for** each story **in** stories **do**
 4:     sentences = SPLIT_SENTENCES(story)
 5:     **for** context_length **from** 3 **to** 10 **do**
 6:         **for** position $i$ **in** valid_range **do**
 7:             context = sentences[i : i + context_length]
 8:             true_next = sentences[i + context_length]
 9:             distractor = RANDOM_SENTENCE(valid_distractor_positions)
10:             option_A, option_B = RANDOMIZE_ORDER(true_next, distractor)
11:             correct_label = 'A' or 'B'
12:             ADD question to all_questions
13:         **end for**
14:     **end for**
15: **end for**
16: WRITE all_questions **to** output_file

---

### A.2  ADDING PERPLEXITY AND SEMANTIC SIMILARITY

To enrich our dataset, we computed two key features for each question's options. Semantic Similarity was calculated by encoding the context and each option into vector embeddings using a SentenceTransformer model and then computing the cosine similarity between the context and option vectors. Perplexity was calculated using powerful, language-specific causal language models. To ensure the perplexity score reflected only the coherence of the option itself, we employed a context masking technique. When calculating the model's loss on the combined context + option text, we set the labels for all context tokens to -100, effectively instructing the model to ignore them. This isolates the perplexity calculation to just the option, conditioned on the context. Refer to the algorithm 2 above.

### A.3  SYSTEM PROMPT FOR PROMPTING MODELS

1. Direct Answering Prompt:

```
def build_prompt(context, opt_a, opt_b):
    return (
        f"Given the following story context:\n\n{context}\n\n"
        "Which sentence comes next?\n\n"
        f"A: {opt_a}\n"
```

```
        f"B: {opt_b}\n"
        "Only reply with a single letter: A or B. Do not say Neither, you
            have to reply with a single letter.")
```

2. Chain of thought (CoT) Prompt:

```
def build_prompt(context, opt_a, opt_b):
```

---

**Algorithm 2** Computing new features

---

**Input:** filePath, llm_model
**Parameter:** sample_size, mask_context
**Output:** output_path of the CSV with new features.
 1: **function** ADDNEWFEATURES(filePath, llm_model, sample_size=10000, mask_context=True)
 2:    df = LOAD_CSV(filePath)
 3:    df = SAMPLE(df, n=sample_size, random_state=42)
 4:    sbert = LOAD_SENTENCE_TRANSFORMER("paraphrase-multilingual-MiniLM-L12-v2")
 5:    device = "cuda" **if** available **else** "cpu"
 6:    tokenizer = LOAD_TOKENIZER(llm_model)
 7:    lm_model = LOAD_LANGUAGE_MODEL(llm_model).to(device)
 8:    df["semantic_sim_A"] = 0.0
 9:    df["semantic_sim_B"] = 0.0
10:    df["ppl_A"] = 0.0
11:    df["ppl_B"] = 0.0
12:    **for** each row **in** df **do**
13:        context = row["context"]
14:        option_A = row["option_A"]
15:        option_B = row["option_B"]
16:        context_embedding = GET_EMBEDDING(context)
17:        df["semantic_sim_A"]          =          COSINE_SIMILARITY(context_embedding, GET_EMBEDDING(option_A))
18:        df["semantic_sim_B"]          =          COSINE_SIMILARITY(context_embedding, GET_EMBEDDING(option_B))
19:        df["ppl_A"] = COMPUTE_PERPLEXITY(context, option_A, mask_context)
20:        df["ppl_B"] = COMPUTE_PERPLEXITY(context, option_B, mask_context)
21:    **end for**
22:    df["delta_semantic_similarity"] = df["semantic_sim_A"] - df["semantic_sim_B"]
23:    df["delta_perplexity"] = df["ppl_B"] - df["ppl_A"]
24:    output_path = GENERATE_OUTPUT_PATH(filePath, mask_context, model_name)
25:    SAVE_CSV(df, output_path)
26: **end function**
27: **function** GET_EMBEDDING(text)
28:    **return** sbert.encode(text, convert_to_tensor=False)
29: **end function**
30: **function** COMPUTE_PERPLEXITY(context, option, mask_context)
31:    full_text = context + " " + option
32:    encoded_tokens = TOKENIZE(full_text, return_tensors="pt").to(device)
33:    **if** mask_context = True **then**
34:        context_length = LENGTH(TOKENIZE(context))
35:        labels = COPY(encoded_tokens)
36:        labels[:, :context_length] = -100
37:    **else**
38:        labels = COPY(encoded_tokens)
39:    **end if**
40:    **with** torch.no_grad():
41:        loss = LANGUAGE_MODEL(encoded_tokens, labels=labels).loss
42:    perplexity = EXPONENTIAL(loss)
43:    **return** perplexity
44: **end function**

---

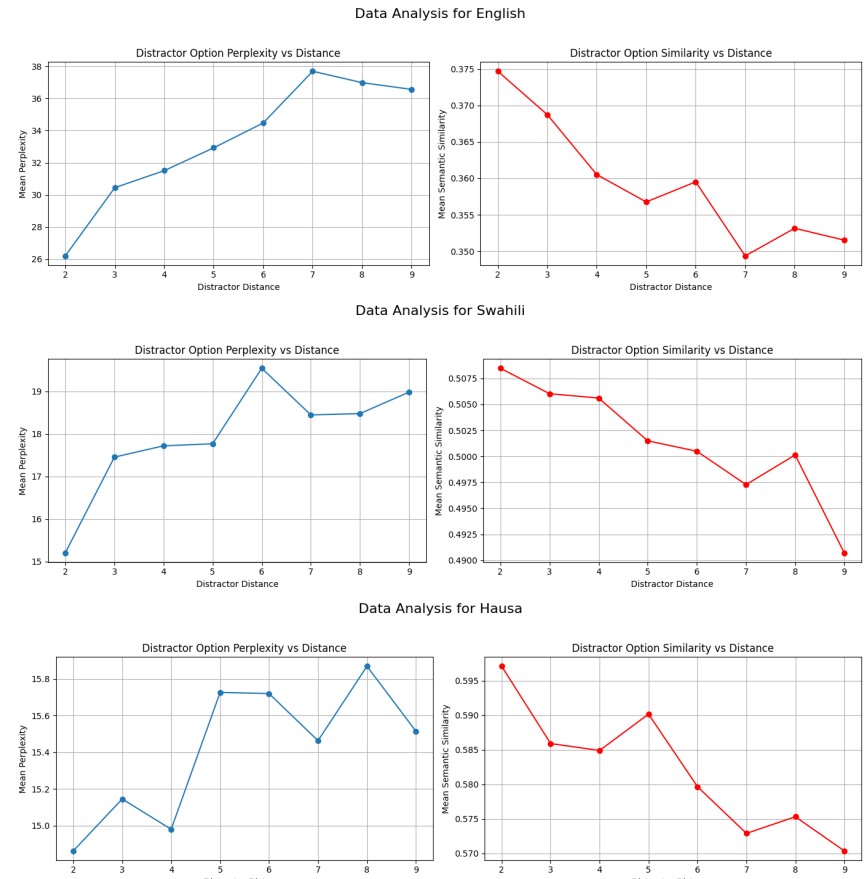

Figure 7: Data Validation (Mean perplexity and similarity vs distractor distance.)

```
return (
    f"Given the following story context:\n\n{context}\n\n"
    "Which sentence comes next?\n\n"
    f"A: {opt_a}\n"
    f"B: {opt_b}\n"
    "Explain your process step by step and end with your final answer
        .")
```

## A.4  VALIDATING DATASET

We confirmed (Figure 7) that as the distractor distance increases, its semantic similarity to the context decreases while its perplexity generally increases.

## A.5  MODEL DECISION BOUNDARY VISUALIZATIONS

The following figures (Figure 8) visualize the decision boundaries for each model on our Hausa dataset, plotting delta_semantic_similarity (x-axis) against delta_perplexity (y-axis). Each point is an NSP question, colored red for incorrect predictions and blue for correct ones. These plots visually confirm the "zone of ambiguity" discussed in Section 5.1.4. The incorrect predictions (red dots) are heavily concentrated around a delta_perplexity of zero, showing that models fail most often when they lack a clear probabilistic signal to guide their choice. The plots also reveal that while GPT-4 and Gemini's errors are tightly clustered around this zone, LLaMA 3's errors are more widely spread out, indicating it struggles under a broader range of conditions.

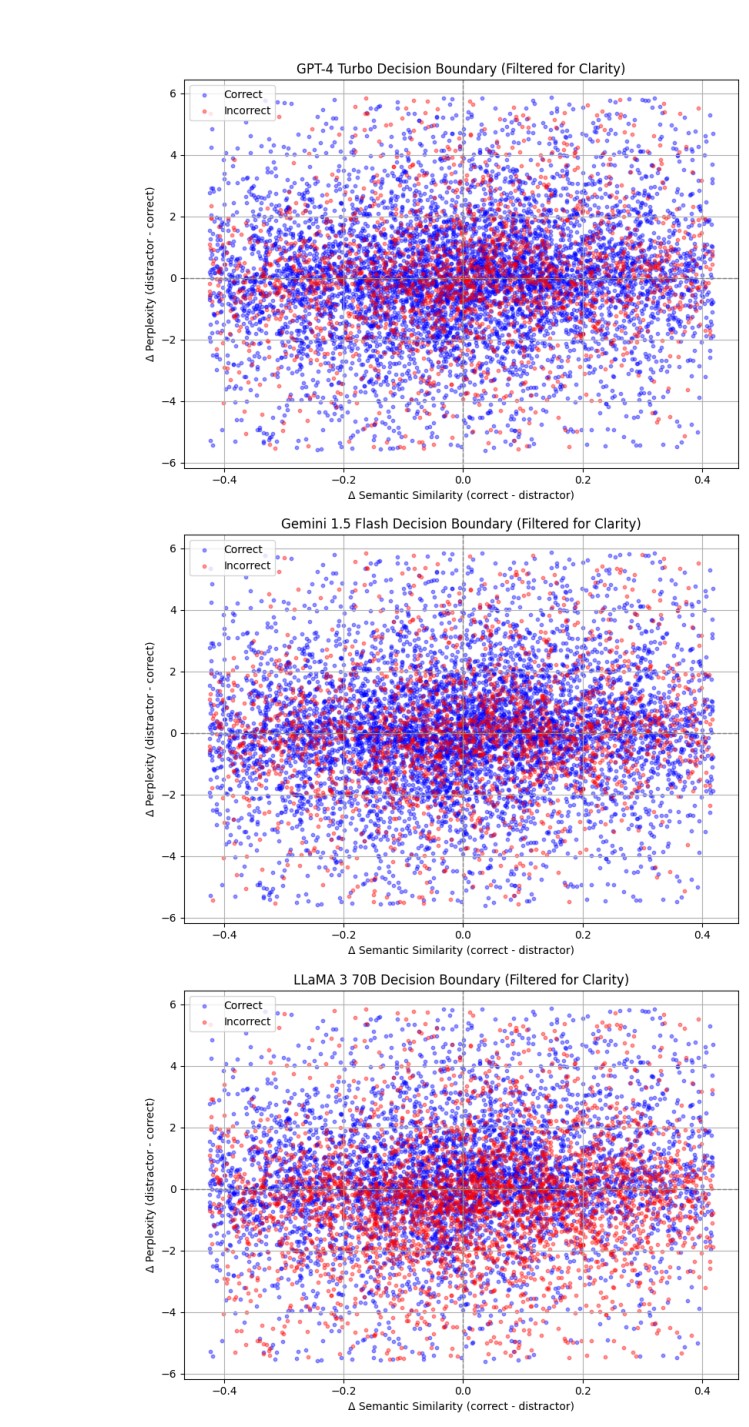

Figure 8: Decision Boundary (Delta perplexity vs Delta Semantic Similarity)

