# OpenReview forum: "Testing Cross-Lingual Text Comprehension in LLMs Using Next Sentence Prediction."
_ICLR.cc/2026/Conference — ICLR 2026 Conference Withdrawn Submission_

### Official Review · Reviewer_sHY8 · 2025-10-26

**Soundness:** 2
**Presentation:** 2
**Contribution:** 1
**Rating:** 2
**Confidence:** 5

**Summary:**

The authors study the Next-Sentence-Prediction capabilities in LLMs (GPT-4-Turbo, Gemini-1.5-Pro, Llama-70b) using a dataset built using publicly available books in 3 languages (English, Swahili, Hausa). They benchmark this ability using vanilla prompting and Chain-of-Thought, and find that CoT might hurt the performance in low-resource languages.

**Strengths:**

- The paper works on 2 important low and medium-resource African languages, Swahili and Hausa, and develops a new benchmark to assess the reading comprehension of LLMs.

**Weaknesses:**

- This paper is not well-written and has too many dramatic adjectives, as if heavily refined by an AI. At the current state, the paper is more like a technical report (more information provided below). It feels like this is an old paper that was resubmitted here.
- Several previous works ([Ahuja _et al._, 2024a](https://aclanthology.org/2023.emnlp-main.258/), [Ahuja _et al._, 2024b](https://aclanthology.org/2024.naacl-long.143/)) have shown and clearly benchmarked the performance of LLMs across various standard tasks, including low-resource African languages, and have shown the performance drops. Also, given that these models are not SOTA anymore (Gemini-2.5, and GPT-5 which claim to be better at multilingual), these findings might not hold.
- Also, the Chain-of-Thought analysis is quite bland and incomplete. During CoT, what language did the model use to reason? It could that that reasoning in a low-resource language could be detrimental. Also, use of "overthinking" is a stretch (for me). Overthinking is generally attributed to spending more tokens on reasoning for a simple problem, and somehow, the word does not sit well with the current experiment design. Given the availability of reasoning models, the authors should have studied their performance, especially while reasoning in English and the query's native language ([Ahuja _et al._, 2025b](https://arxiv.org/abs/2507.00246)). I feel some analysis of the CoT trace should also be conducted, i.e., the number of tokens on easier and harder classification problems.
- As mentioned before, the study is bland and just revolves around benchmarking models on a single classification task, NSP, and contains some unnecessary technical details. The prompt used for CoT is also a very rudimentary one-liner, and I am not sure about its efficacy.
- No mitigation strategies or improvement methodologies are discussed.
- Another interesting analysis would be pre/post CoT, i.e., _reason-and-answer_ (vs) _answer-and-reason_. This woudl motivate how the LLM can "fix and justify" a CoT to a wrong answer as well.
- While predicting A/B, no option-bias is analysed, which is important for classification tasks with LLMs.
- I believe this paper requires a much deeper analysis and problem motivation for ICLR. I believe this work is superseded by the large-scale benchmarking studies and works like [Ahuja 2025a](https://arxiv.org/abs/2504.11900).

**Questions:**

- Was sampling disabled for these experiments?
- While scraping the data, was the license of the books respected?

---

### Official Review · Reviewer_raRS · 2025-10-26

**Soundness:** 1
**Presentation:** 1
**Contribution:** 1
**Rating:** 2
**Confidence:** 5

**Summary:**

This paper presents a next sentence prediction (NSP) evaluation dataset with data in 3 languages (English, Hausa, Swahili). The authors argue this task (NSP) provides a new way to evaluate a model's conceptual understanding of a narrative. This dataset is then used to evaluate three LLMs and compare them to a number of heuristic features. These heuristic features each have a strong relationship with the results. Then, CoT prompting is evaluated and results in very inconsistent impact on this benchmark.

**Strengths:**

1. the paper has clear motivation and presentation. It is easy to read and understand.
2. the paper provides sufficient details for reproducibility

**Weaknesses:**

*Major*:
1. The paper displays little awareness of the field it seeks to contribute to (multilingual NLP or NLP in general).
- related works section doesn't discuss any prior work on evaluating multilingual text comprehension
- paper doesn't discuss any prior work on multilingual CoT prompting
- paper never compares this method and dataset to alternatives, never frames results with respect to other recent literature, and never discusses linguistic features that could relate.

As a result, given the complete lack of contextualization, the contributions of this paper are very unconvincing.

*Other*:
1. NSP was abandoned in the field of NLP as models got better (as the authors mention in the related works paper, citing the RoBERTa). The paper fails to display why NSP would be a useful test (relative to all the evaluations available) with the tremendous advancements of LLMs. The paper shows the relationship with surface-level features (e.g. distractor length) as a finding, which is exactly why NSP is no longer useful. As a reviewer, I am unconvinced that NSP = comprehension.
1. Perplexity analysis is quite unsound. Not only, is perplexity evaluated using smaller/weaker models, but multiple of them are Llama-3 based. Authors fail to address how strongly related perplexity (i.e. next token prediction) is to next sentence prediction (especially within same model family) and
1. Comparison across languages with different data. The paper does cross-lingual comparisons of performance, but the dataset splits were built on individual texts, meaning there's no validation that, for example, the Hausa storybooks are simply much more difficult to do NSP on than the other languages. In addition, linguistic considerations are not mentioned. Could NSP be fundamentally more difficult in different languages.
1. Analysis shows correlations, but doesn't run any ablations to display a higher level of relationship (e.g. causality).
1. Lack of statistical details (variance, error bars, stat sig). Notably, the CoT results don't seem stat sig given just 1000 questions with only two choices.

*Minutia*:
1. L80 "While vital for measuring AI’s progress, these benchmarks are often
too complex for simple and scalable use in practical applications like educational apps."
1. L87 "superficial cues" ?
1. L302 why is this surprising ?
1. no evidence
1. lots of overstated language / unsupported claims across the paper (L14, L53, L139, L262, L356, L427, to name a few)
1. informal language across the paper

**Questions:**

Questions:
* Is this work targeting the education space ? If so, clearer links should be made. If not, the recurring mentions of education seem out of place. What is RoboTutor ?

---

### Official Review · Reviewer_1mxT · 2025-11-01

**Soundness:** 1
**Presentation:** 3
**Contribution:** 1
**Rating:** 2
**Confidence:** 3

**Summary:**

This paper proposes a new cross-lingual benchmark to evaluate text comprehension in large language models (LLMs) using a Next Sentence Prediction (NSP) task. The authors constructed a balanced dataset of 10,000 questions for English, Swahili, and Hausa. They evaluated three high-performance models (GPT-4 Turbo, Gemini 1.5 Flash, and LLaMA 3 70B) and analyzed their performance with and without Chain-of-Thought (CoT) prompting. The authors claim that model accuracy degrades significantly in low-resource settings and that CoT prompting hurts stronger models due to "overthinking." However, the study's conclusions about "genuine comprehension" and "reasoning" are built on a methodologically fragile foundation, and the evidence presented is insufficient to support its primary claims.

**Strengths:**

The creation of a large, balanced, three-language dataset sourced from African Storybooks is a notable contribution to the community. This resource could be valuable for future research, if the number of languages would be increased further.

**Weaknesses:**

Task Misalignment: NSP is an inadequate proxy for “comprehension” or “reasoning”. Its reliance on shallow heuristics invalidates the core claims.

Lack of Statistical Rigor: No significance testing, confidence intervals, or regression coefficients are reported. Claims of “most influential factors” and CoT effects are unsupported.

No Human Baseline: Without human performance, the reported accuracies are uninterpretable.

Limited Model Diversity: Only three large models are tested, leading to overgeneralized conclusions about “stronger” vs. “weaker” systems.

Overstated Conclusions: The findings likely reflect heuristic transfer, not true cross-lingual reasoning.

Missing important/relevant references:
https://arxiv.org/abs/2505.14395
https://aclanthology.org/2025.naacl-long.139/
https://aclanthology.org/2024.sigtyp-1.14/
https://arxiv.org/abs/2412.00948

**Questions:**

How do you justify using Next Sentence Prediction (NSP) as evidence of “comprehension” or “reasoning”?

Are the reported differences in accuracy and “most influential factors” statistically significant? Please provide confidence intervals or test results to substantiate these claims.

What is human performance on your benchmark, and how does it contextualize the reported model accuracies?

Model Scope: Why were only three large proprietary models evaluated?

---

### Official Review · Reviewer_dJXM · 2025-11-09

**Soundness:** 3
**Presentation:** 2
**Contribution:** 2
**Rating:** 4
**Confidence:** 4

**Summary:**

This paper builds a large, balanced cross-lingual benchmark for Next Sentence Prediction (NSP) with 10,000 items each in English (high-resource), Swahili (medium-resource), and Hausa (low-resource). It evaluates GPT-4 Turbo, Gemini 1.5 Flash, and LLaMA-3-70B, and studies factors such as distractor distance/length, semantic similarity, and (masked) perplexity. Key findings: accuracy degrades as language resources decrease; Chain-of-Thought (CoT) helps the weakest model (LLaMA-3) but often hurts stronger models (GPT-4/Gemini) in lower-resource languages; and models fail most when “delta” similarity/perplexity between options is near zero (a “zone of ambiguity”). The paper also proposes using CoT explanations pedagogically.

Overall, I think this could be a useful benchmark/analysis paper. With several clarifications and stronger experimental controls (detailed below), I’d be willing to raise my score.

**Strengths:**

1. Balanced, large-scale cross-lingual NSP (10k per language) sourced from African Storybooks with a clear generation pipeline.
2. Thoughtful analyses (logistic regression feature importance; ambiguity via delta-PPL and delta-similarity).

**Weaknesses:**

1. The paper states it is the first to systematically apply/analyze CoT for cross-lingual NSP and proposes pedagogical use. Please situate this claim relative to prior CoT-style multilingual comprehension work and clarify what “systematically” entails (e.g., prompt design/search, coverage across languages/models).

2. NSP can be susceptible to shallow cues. The paper mitigates this with engineered features and analyses, but please discuss residual artifacts that NSP may still capture vs. “true” discourse comprehension—especially across languages with different morphology/length distributions.

**Questions:**

1. Table 3 reports the “single most influential factor.” Was this from single-feature logistic models or a multivariate model with standardized coefficients? If single-feature, consider providing multivariate results or SHAP-style analyses to avoid confounds between distance/length/similarity.

Minor Comments / Typos
1. The section/TOC listing at the end seems to duplicate “Analysis of Factors Influencing Errors.” Consider deduplicating/renaming subsections (e.g., 5.1 vs 5.1.1).

---

### Note · Authors · 2025-11-21

I have read and agree with the venue's withdrawal policy on behalf of myself and my co-authors.